# β-Cardiac myosin hypertrophic cardiomyopathy mutations release sequestered heads and increase enzymatic activity

Arjun S. Adhikari [1,2], Darshan V. Trivedi[1,2], Saswata S. Sarkar [1,2], Dan Song[1,2], Kristina B. Kooiker[1,2,3], Daniel Bernstein [2,3], James A. Spudich [1,2] & Kathleen M. Ruppel [1,2,3]

Hypertrophic cardiomyopathy (HCM) affects 1 in 500 people and leads to hyper-contractility of the heart. Nearly 40 percent of HCM-causing mutations are found in human β-cardiac myosin. Previous studies looking at the effect of HCM mutations on the force, velocity and ATPase activity of the catalytic domain of human β-cardiac myosin have not shown clear trends leading to hypercontractility at the molecular scale. Here we present functional data showing that four separate HCM mutations located at the myosin head-tail (R249Q, H251N) and head-head (D382Y, R719W) interfaces of a folded-back sequestered state referred to as the interacting heads motif (IHM) lead to a significant increase in the number of heads functionally accessible for interaction with actin. These results provide evidence that HCM mutations can modulate myosin activity by disrupting intramolecular interactions within the proposed sequestered state, which could lead to hypercontractility at the molecular level.

[1] Department of Biochemistry, Stanford University School of Medicine, Stanford, CA 94305, USA. [2] Stanford Cardiovascular Institute, Stanford, CA 94305, USA. [3] Department of Pediatrics (Cardiology), Stanford University School of Medicine, Stanford, CA 94305, USA. Correspondence and requests for materials should be addressed to J.A.S. (email: jspudich@stanford.edu) or to K.M.R. (email: kruppel@stanford.edu)

Hypertrophic cardiomyopathy (HCM) is a heritable cardiovascular disorder characterized by abnormal thickening of the left ventricular walls[1], preserved or increased systolic function and reduced diastolic function. HCM is most often caused by mutations in genes encoding sarcomeric proteins[2], principally those encoding β-cardiac myosin (*MYH7*) and cardiac myosin binding protein-C (*MYBP3*)[3,4]. HCM is typically diagnosed in late adolescence or adulthood, and is the leading cause of sudden cardiac death in those under age 35[5]. Current treatment for HCM is limited to symptomatic relief, and includes heart muscle reduction surgery (myectomy), defibrillator placement, and even heart transplant in the most refractory cases. There is an urgent need for new therapeutic approaches to the disease, but first we need to fully understand its underlying molecular basis.

It has been hypothesized that mutations in β-cardiac myosin, the mechanoenzyme that drives ventricular contraction, cause HCM by affecting the power output of the myosin motor[6]. β-cardiac myosin is a hexamer consisting of two heavy chains, two essential light chains (ELC) and two regulatory light chains (RLC). The heavy chains can be further divided into heavy meromyosin (HMM) and light meromyosin (LMM). LMM is the distal tail of myosin and contains the sequences responsible for filament formation. HMM is comprised of Subfragment 1 (S1), the head or motor domain of myosin, and Subfragment 2 (S2), the first ~40% of the α-helical coiled-coil tail.

Previous studies of the effects of HCM mutations on myosin's biomechanical parameters at the molecular level have shown variable results. Studies using myosin derived from mouse cardiac muscle showed significant increases in ATPase activity, actin gliding velocity, and intrinsic force[7,8]. However, mouse hearts contain predominantly α-cardiac myosin, which differs from human β-cardiac myosin by ~80 residues in the catalytic domain alone. Subsequent studies performed on the R403Q HCM mutation engineered into either the mouse α-cardiac or β-cardiac myosin backbone showed significant differences in the biomechanical effects of the mutation depending on the isoform into which it was introduced[9]. Moreover, investigations of the effects of HCM mutations on the human β-cardiac myosin motor domain containing the essential light chain (MDE; referred herein as short S1, or sS1), have shown that there are significant differences between the ATPase activities and actin gliding velocities

of purified recombinant human sS1 α- and β-cardiac myosins[10]. Experiments performed with proteins isolated from human biopsy samples have also produced conflicting results, perhaps in part because these samples are a mixture of human WT β-cardiac, mutant β-cardiac, and α-cardiac myosins[11–13]. These results show that to understand how HCM mutations alter human β-cardiac myosin function at the molecular level, one needs to study the disease using a human β-cardiac myosin backbone. This has been made possible by the development of a mouse myoblast expression system[14–16]. Using this system, we have investigated the effect of several HCM mutations in the catalytic head domain of human β-cardiac myosin on ATPase activity, actin gliding velocity, and intrinsic force[17–20]. We have seen in most cases (except for early onset HCM mutations[17]) that there are mostly only very small changes (~10–20%) compared to WT sS1[18–20], and that not all parameter changes are a gain in function. A notable example is R719W, a clinically pathogenic mutation used in the present study, which results in ~10% decrease in the intrinsic force of human β-cardiac sS1[20]. Thus, in many cases one cannot explain the clinical hypercontractility seen in HCM patients by changes in these fundamental biomechanical properties of myosin.

Recent work has emphasized the existence of a relatively flat, arginine-rich, and positively charged surface on the myosin catalytic head, termed the myosin mesa[21,22]. It was proposed that the myosin mesa may act as a binding interface with MyBP-C[21] or the proximal part of S2[23], sequestering myosin heads into a state that cannot interact with actin. A possible unifying hypothesis for HCM mutant-induced hypercontractility posited that many, if not most, HCM mutations result in the release of heads from this putative sequestered state, causing an increase in the number of myosin heads available for interaction with actin. There is now considerable support for this hypothesis[17,21–28]. While the high-resolution structure of such a sequestered form of human β-cardiac myosin heads is yet to be determined, the best working structural hypothesis involves a known folded-back sequestered structure involving interactions between different subdomains of myosin referred to as the interacting heads motif (IHM; Fig. 1a)[29–36]. The IHM structure has an asymmetric conformation of the two S1 heads, one head thought to be blocked for actin interaction (the blocked head) and one which possibly could still interact with actin (the free head). There are low-resolution

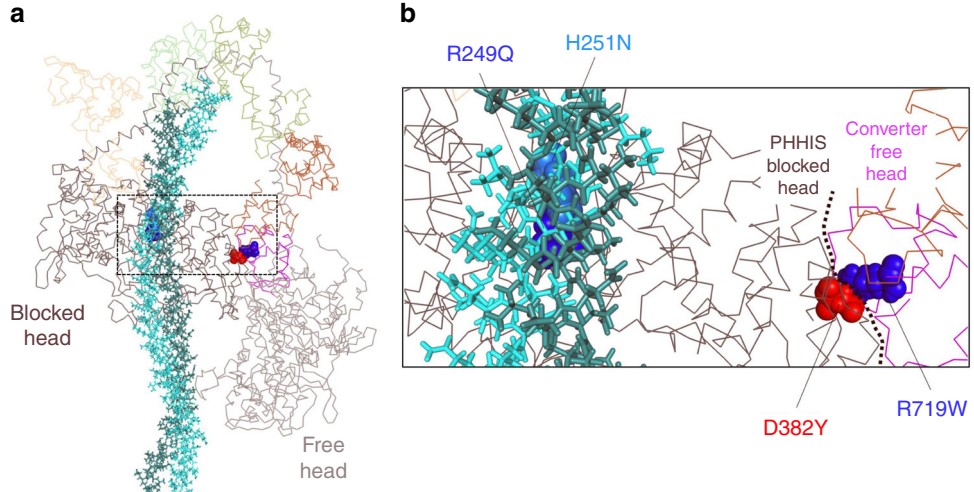

**Fig. 1** Location of mutated residues on folded back model of human β-cardiac myosin. **a** Model of the back-side view of the human β-cardiac myosin IHM (MS03 homology model[26]), with the alpha-carbon backbones of the two S1 heads and the light chains shown as lines, and the S2 tail region represented by sticks. The four residues mutated in this study are represented by spheres. **b** Close up of region outlined by the dashed box in figure a showing the mutated residues R249Q (blue) and H251N (light blue) at the head-tail interface and D382Y (red) and R719W (blue) at the head–head interface

(~2 nm) structures of this IHM state for multiple myosin types[31,34,36–40], including human β-cardiac myosin[29], and it seems likely that something close to the folded-back structure shown in Fig. 1 is the sequestered state of interest. The data reported here support this structural hypothesis.

Homology models of human β-cardiac myosin have shown that in the myosin IHM state, the myosin mesa is positioned to interact with the proximal part of the S2 tail[22,25]. Moreover, the models depict a major site of interaction of the two S1 heads with one another in the IHM folded state that involves a primary head–head interaction site (PHHIS) on the blocked head interacting with the converter domain of the free head (Fig. 1b). Consistent with this structural hypothesis, the myosin mesa[17,22,25,26], the proximal region of S2[22] and the converter domain[20,22] are three known hot spots for myosin HCM mutations.

Recently, the HCM mutations R453C, R249Q, and H251N on the myosin mesa have been shown to weaken the S1-proximal S2 interaction[17,25], consistent with the structural model (see, for example, the positions of R249Q and H251N in Fig. 1b) and with the unifying hypothesis that release of sequestered heads due to HCM mutations lead to an increase in the number of myosin heads accessible for interaction with actin, thus leading to hypercontractility. Importantly, while these data showed that the HCM mutations disrupt binding between the two myosin domains, they did not show that this disruption of binding actually leads to increased myosin activity.

To obtain functional data to test the unifying hypothesis, we used two previously described recombinant human β-cardiac myosin HMM-like two-headed molecules[25], which differ with respect to the number of proximal S2 heptad repeat units they contain. The first construct (25-hep HMM) consists of two human β-cardiac myosin S1 heads, the human cardiac essential and regulatory light chains (ELC and RLC), and 25 heptad repeats of the proximal S2 tail. This construct consists of all the components required for the myosin heads to fold back into the putative IHM state, and it has been shown that the ATPase activity of this construct is regulated by RLC phosphorylation[25]. The second HMM-like construct (2-hep myosin) consists of two human β-cardiac myosin S1 heads, the human cardiac essential and regulatory light chains (ELC and RLC), but only the first 2 heptad repeats of the proximal S2 tail, and is not regulated by RLC phosphorylation[25]. Each construct has a carboxyterminal GCN4 leucine zipper domain to ensure dimerization. We introduced four different HCM-causing missense mutations into the 2- and 25-hep HMM constructs in parallel. Two mutations, R249Q and H251N, affect residues on the myosin mesa at the S1–S2 interface (Fig. 1), and were previously shown to affect binding between S1 and S2[17,25]). The other two mutations, D382Y and R719W, affect residues at the modeled PHHIS-converter interface (Fig. 1).

First, we probe the fraction of myosin heads that are in the super-relaxed state (SRX)[41–43], defined as a state of myosin with an extremely low ATPase turnover rate. We use mant-ATP single turnover experiments to calculate the fraction of myosin heads hydrolyzing ATP at their normal basal (~0.03 s$^{-1}$) rate versus their SRX (~0.003 s$^{-1}$) rate[24,41,42]. A recent publication by Anderson et al.[24] showed that the SRX corresponds to a folded-back (sequestered) state of myosin, possibly the IHM described previously[30,31]. Our studies show that the mutant 25-hep HMMs show a much greater fraction of myosin heads in the fast phase compared to WT 25-hep HMM, consistent with our hypothesis that these mutations disrupt the myosin intramolecular interactions that sequester heads in the SRX inactive state.

We also use actin-activated ATPase assays to show that these mutations indeed lead to increased ATPase activity of the mutant

25-hep HMMs compared to the WT 25-hep HMMs. These two experimental approaches show that mutations at the S1–S2 and S1–S1 interfaces of the IHM model structure lead to increased catalytic activity at the molecular level, and support our hypothesis that these mutations cause more heads to become accessible for interaction with actin. Moreover, the results suggest that when studying HCM mutations in the human β-cardiac myosin backbone, it is not sufficient to study just the head domain, but the HMM must also be investigated to account for how these mutations may alter the intramolecular interactions involving the two heads with one another and with their S2 tail.

## Results

**Selection of four HCM mutations located at IHM interfaces.** Our primary aim was to investigate how HCM-causing mutations located within regions modeled to be important for either S1–S2 or S1–S1 interactions in the IHM state alter myosin enzymatic activity. We chose four mutations, all of which are known to be pathogenic clinically. To study the S1–S2 region interaction, we selected two mutations on the myosin mesa, R249Q and H251N, that are in a position to interact with proximal S2 (Fig. 1). Previous work has shown that these mutations weaken S1–S2 binding[17,25]. To study the change in myosin contractility due to mutations at the PHHIS-converter S1–S1 interface, we chose R719W and D382Y (Fig. 1).

Microscale thermophoresis (MST) is a technique that follows the diffusion of a fluorescent probe along a temperature gradient and can be used to measure biomolecular interactions over a broad range of affinities, including a $K_D$ in the double-digit μM range. Using MST, we previously showed that the R249Q, H251N, and R453C mesa mutations, which are in close proximity to proximal S2 in the IHM model (see, for example, Fig. 1), cause large decreases in the affinity of sS1 or 2-hep HMM for proximal S2 (R453C, >5 fold; R249Q, >6 fold; and H251N, >4 fold)[17,25]. Here, we used MST to investigate whether the R719W and D382Y mutations alter the interaction between 2-hep HMM and proximal S2. According to the IHM model, they would not be expected to do so, unless perhaps by some long-distance allosteric effect (Fig. 1). Both R719W (58 ± 4 μM; mean + SEM for two biological replicates with three technical replicates) and D382Y (45 ± 12 μM) 2-hep HMM bind to proximal S2 with slightly weaker affinities than that of WT 2-hep myosin (37 ± 7 μM) (Fig. 2), but the observed affinities represented <2-fold change compared to the WT 2-hep HMM-proximal S2 affinity.

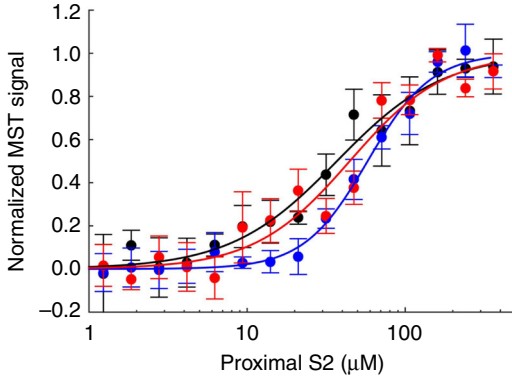

**Fig. 2** Mutations R719W and D382Y do not affect proximal S2 binding. Microscale thermophoresis of WT (black), R719W (blue), and D382Y (red) 2-hep HMM titrated with proximal S2. Representative data from one preparation of myosin, and three experimental repeats. Error bars represent the SEM. Source data are provided as a Source Data file

While these binding studies are consistent with both the structural model and the unifying hypothesis, functional studies are needed to test whether mutations located on these important myosin binding surfaces indeed alter myosin activity. We used functional assays to explore both 2-hep and 25-hep HMMs containing the HCM mutations, with the 2-hep HMM serving as a control for maximal activity. We used the 2-hep HMM as a control because it shows the same maximal actin-activated ATPase activity as sS1 alone (Supplementary Figs. 1 and 2), presumably because of the lack of a proximal S2 tail for the heads to fold back onto. Moreover, using the 2-hep HMM as a control for the 25-hep HMM (each with the same mutation), allowed us to specifically probe how the mutation alters the interaction between different myosin domains, rather than how the mutation alters the catalytic activity of the myosin S1 head.

**HCM mutations on myosin mesa increase 25-hep HMM activity.** Anderson et al. showed that the level of SRX measured by mant-ATP single turnover by WT 25-hep HMM corresponds well to a folded-back state of the 25-hep WT HMM, as observed by negative staining electron microscopy[24]. To investigate whether the HCM mutations R249Q and H251N release sequestered myosin heads into a functionally active state, we used mant-ATP single turnover kinetics, a technique originally used to probe the level of SRX in cardiac myosin fibers[41,42] which has more recently been applied to purified cardiac myosin[24,44]. When bound to myosin, mant-ATP shows increased fluorescence. When chased with unlabeled ATP, the release of mant-nucleotide from the myosin can be monitored by a decrease in fluorescence over time (Fig. 3a). If the myosin is in an open state, then its mant-nucleotide release rate in the absence of actin is higher ($\sim$0.03 s$^{-1}$; fast) than that of the sequestered SRX ($\sim$0.003 s$^{-1}$; slow)[30,41]. Fitting the mant-ATP fluorescence vs time data to a double exponential [Eq. (1)) gives the fraction of myosin in the SRX slow phase.

$$F = a_1 \times \exp(b_1 \times t) + a_2 \times \exp(b_2 \times t) \qquad (1)$$

where $F$ is fluorescence (arbitrary units), $a_1$ and $a_2$ are the fraction of myosin in the slow and the fast phase, and $b_1$ and $b_2$ are the slow and fast rates, respectively.

For all 2-hep HMMs (WT and mutants) $\sim$80% of the myosin heads show fast-phase kinetics (Figs. 3–7 and Supplementary Tables 2 and 3), very similar to that found for sS1[24]. For WT 25-hep HMM, however, $41 \pm 7\%$ (SEM for two biological replicates, with three technical replicates) was in the fast phase, compared to $81 \pm 3\%$ for the WT 2-hep HMM (Fig. 3, Supplementary Tables 2 and 4). This significant decrease in the fast phase suggests that some of the heads in the 25-hep HMM construct are likely in a folded sequestered state due to the presence of the S2 tail.

However, for H251N and R249Q 25-hep HMM, a much higher fraction of the myosin heads, $65 \pm 4\%$ and $79 \pm 3\%$, showed fast-phase kinetics (Fig. 4a, c, Supplementary Tables 2 and 4), corresponding to a 41% and 59% decrease in the number of SRX heads, respectively. These data are consistent with HCM mutations at the S1–S2 interface disrupting the myosin IHM sequestered state.

To confirm these functional effects of the HCM mutations on myosin catalytic function, we compared the actin-activated ATPase activities of the 2-hep and 25-hep HMMs containing the R249Q and H251N mutations, with the 2-hep HMM of each mutant serving as a control for maximal activity. Previous experiments have shown that the ratio between the $k_{cat}$ (derived from the Michaelis–Menten equation) of 25-hep HMM to 2-hep HMM is $0.57 \pm 0.03$ (mean + SEM for two biological replicates, two technical replicates) for unphosphorylated WT human $\beta$-cardiac myosin (Fig. 3b, Supplementary Table 1), consistent

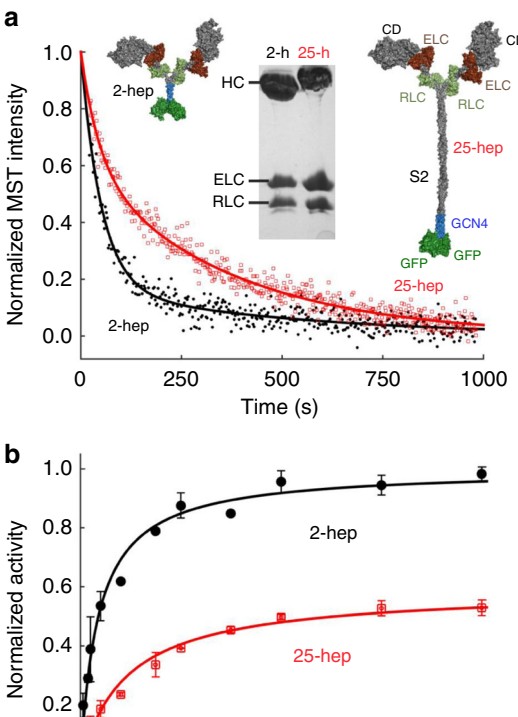

**Fig. 3** WT 25-hep HMM has lower ATPase activity than WT 2-hep HMM. **a** Mant-ATP single turnover kinetics of WT 2-hep HMM (black solid line and filled circles) and 25-hep HMM (red solid line and empty squares). Representative data from one preparation of each protein. Data is fit to a double exponential [Eq. (1)), with the slow phase in the 25-hep HMM clearly visible. Inset shows models of the 2-hep and 25-hep HMM, along with an SDS-PAGE gel showing the human heavy chain, the human ELC and the human RLC for both the 2-hep and 25-hep HMM (for uncropped image, see Supplementary Fig. 4). **b** Actin-activated ATPase activity of WT 2-hep HMM (black solid line and filled circles) and WT 25-hep HMM (red solid line and empty squares). Data is combined from two independent protein preparations with two experimental replicates for each preparation. Each point is an average with the error bar representing the SEM. Source data are provided as a Source Data file

with about 42% of the 25-hep HMM heads being in a folded-back sequestered SRX state.

Performing the same experiment on the unphosphorylated mutant HMMs, we observed that the 25-hep HMM $k_{cat}$: 2-hep HMM $k_{cat}$ ratio was $0.96 \pm 0.06$ (mean + SEM for three biological replicates with three technical replicates; Fig. 4b, Supplementary Table 1) for R249Q HMM and $0.91 \pm 0.03$ (Fig. 4d) for H251N HMM. These $k_{cat}$ ratios for both mutations were much higher than that of the WT HMM ($0.57 \pm 0.03$), and correspond to a conversion of 91% and 79% of the SRX heads normally in the WT 25-hep HMM under these conditions into functionally accessible heads for interaction with actin.

Note that all of the above experiments used unphosphorylated HMMs (the human RLC was expressed in E. coli and exchanged onto the HMM and was therefore in a fully dephosphorylated state). Phosphorylation of the RLC shifts the equilibrium to the open state and increases the ratio of 25-hep $k_{cat}$: 2-hep $k_{cat}$ to 1 (see Nag et al.[25]), and presumably is the normal physiological effector controlling the number of myosin heads in a functional form.

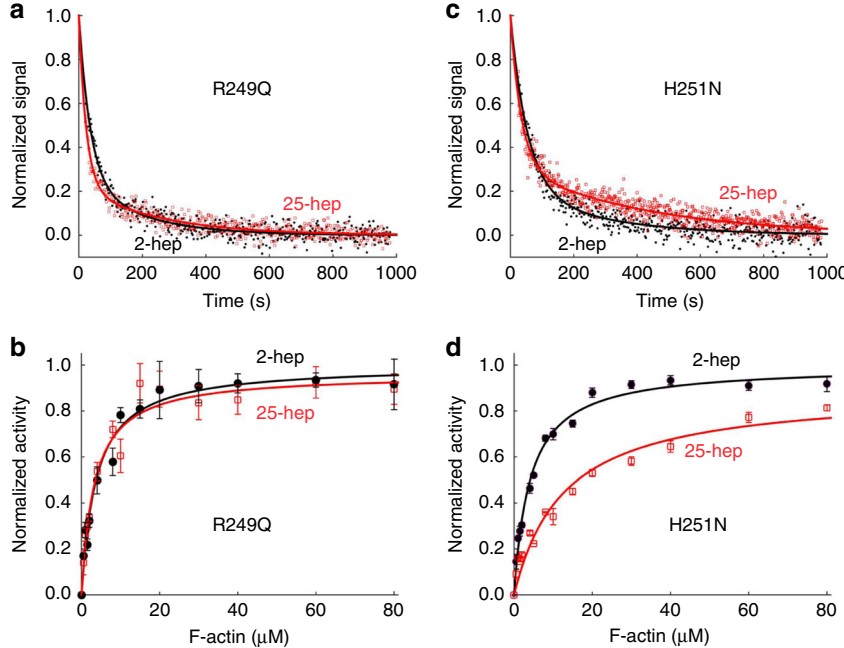

**Fig. 4** Effect of mesa mutations R249Q and H251N on HMM ATPase kinetics. In all cases, black solid line and filled circles are 2-hep HMM data; red solid line and empty squares are 25-hep HMM data. **a** Mant-nucleotide single turnover kinetics for R249Q 2-hep HMM and 25-hep HMM. **b** Actin-activated ATPase data for R249Q 2-hep and 25-hep HMM. **c** Mant-nucleotide single turnover kinetics for H251N 2-hep HMM and 25-hep HMM. **d** Actin-activated ATPase data for H251N 2-hep and 25-hep HMM. **a**, **c** show representative data from one preparation each. Each of panels **b** and **d** are combined data from three independent preparations of HMMs with three experimental replicates for each preparation. Each point is an average with the error bar representing the SEM. Source data are provided as a Source Data file

**Mutations at S1-S1 interface increase 25-hep HMM activity.** Similar to the mesa mutations, the two HCM mutations at the PHHIS-converter S1-S1 interface, R719W and D382Y, cause a decrease in the level of SRX heads. For R719W and D382Y 25-hep HMM, we observed $81 \pm 7\%$ and $68 \pm 4\%$ showing fast-phase kinetics, respectively (Fig. 5a, c, Supplementary Tables 2 and 4). These numbers are significantly higher than the fraction of WT 25-hep HMM myosin heads in the open state ($41 \pm 7\%$) and correspond to a conversion of 68% and 46% of the SRX heads normally in the WT 25-hep HMM under these conditions into functionally accessible heads for interaction with actin.

Consistent with the mant-ATP results, using the actin-activated ATPase assay we observed that the activities of the mutant 25-hep HMMs were close to that of their corresponding 2-hep HMMs. The 25-hep $k_{cat}$: 2-hep $k_{cat}$ ratios were $1.03 \pm 0.03$ for R719W HMM and $0.84 \pm 0.04$ for D382Y HMM, respectively (mean + SEM for 2 biological replicates with 3 technical replicates; Fig. 5b, d, Supplementary Table 1). These changes correspond to a conversion of 100% and 63% of the SRX heads normally in the WT 25-hep HMM under these conditions into functionally accessible heads for interaction with actin.

Thus, the combination of results from the mant-ATP single-turnover and actin-activated ATPase experiments show that these mutations at the PHHIS-converter S1-S1 interface, a known hotspot for HCM mutations, lead to increased 25-hep HMM activity compared to WT 25-hep HMM, presumably by releasing heads from the closed sequestered IHM state. Since the ATPase activities of 2-hep HMM, which lacks the S2 tail, are not altered by these mutations (relative to sS1 mutant controls), these results are consistent with the idea that the myosin PHHIS-converter S1-S1 interaction depends on having the ability to fold back onto proximal S2.

**Net charge-charge interactions involved at S1–S1 interface.** Next, we sought to better understand what types of interactions are occurring between the two heads at the PHHIS-converter S1–S1 interface. In the homology model, the S1–S2 interaction involves the arginine-rich positively charged mesa interacting with the negatively charged proximal S2 tail. In the S1–S1 interaction, we considered two possible charge–charge interaction types. (1) Based on the homology model, the Arg719 residue could specifically interact with the Asp382 residue forming a salt bridge (Fig. 6c), or (2) there is a net positive charge on the free head converter surface, which interacts with an overall negatively charged surface on the blocked head (Fig. 6c–e). To distinguish between these two possibilities, we engineered a D382R/R719D double mutant, where the charges were switched but a salt bridge could possibly still form. According to interaction type 1, this double mutant might be expected to behave like WT 25-hep HMM. According to interaction type 2, this double mutant might be expected to be at least as disruptive of the sequestered state as the individual R719W and D382Y mutations.

In both the mant-ATP single turnover experiment and the actin-activated ATPase assay, the 25-hep HMM double mutant behaved very similar to the 2-hep HMM double mutant (Fig. 6a, b, Supplementary Tables 1–4), which was essentially the same as the WT 2-hep HMM. This suggests that the S1–S1 interaction is disrupted by the double mutation, consistent with a general charge–charge interaction, i.e., interaction type 2 (Fig. 6c–e). However, simulations performed by the Houdusse lab[28] show that R719 interacts with the top loop of the converter domain, which is predicted to make up part of the PHHIS-converter interface. In their simulations, the tryptophan mutation at position 719 destabilizes the interaction with the top loop and decreases the internal dynamics of the converter and converter/ELC interface. Thus, we cannot exclude that the effects of the R719W mutation and the D382R/R719D double mutation on

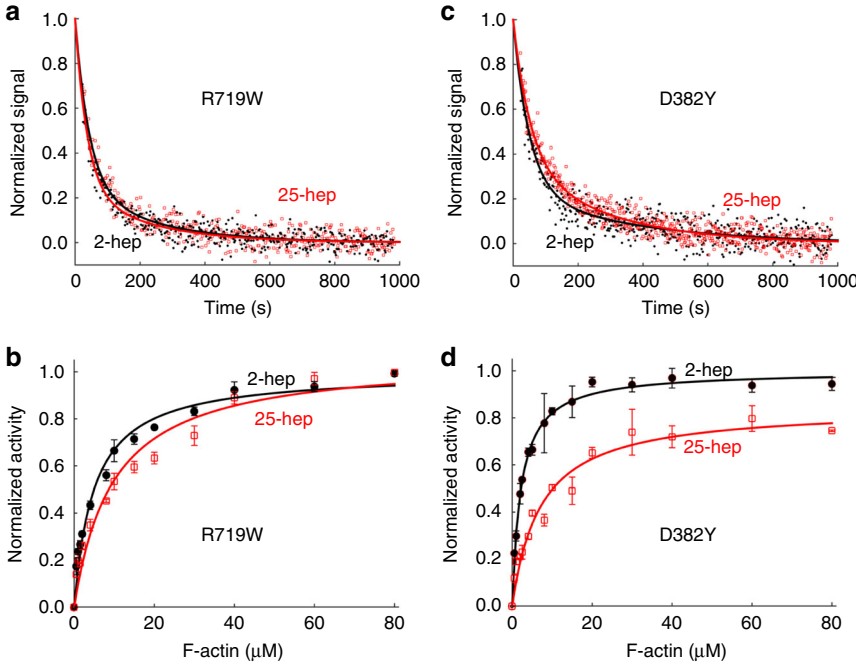

**Fig. 5** Effect of PHHIS-converter domain interaction mutations R719W and D382Y on HMM ATPase kinetics. In all cases, black solid line and filled circles are 2-hep HMM data; red solid line and empty squares are 25-hep HMM data. **a** Mant-nucleotide single turnover kinetics for R719W 2-hep and 25-hep HMM. **b** Actin-activated ATPase data for R719W 2-hep and 25-hep HMM. **c** Mant-nucleotide single turnover kinetics for D382Y 2-hep and 25-hep HMM. **d** Actin-activated ATPase data for D382Y 2-hep and 25-hep HMM. **a**, **c** show representative data from one preparation each. Each of panels **b** and **d** are combined data from two independent preparations of HMMs with three experimental replicates for each preparation. Each point is an average with the error bar representing the SEM. Source data are provided as a Source Data file

the PHHIS-converter S1-S1 interface stem from changes in the conformation of the converter domain of the free head rather than from the loss of the positively charged arginine side chain affecting interaction with the negatively charged surface of the blocked head directly.

**I457T has little effect on the SRX state of β-cardiac myosin.** As a control, we studied a mutation, I457T, which is neither at the PHHIS-converter S1–S1 nor the S1–S2 interface of the human β-cardiac IHM homology model (Fig. 7a, b). I457T is a likely pathogenic mutation[27] and is found in the transducer region of myosin. A recent modeling study by Alamo et al.[27] suggested that this mutation is in a position that should not disrupt the myosin IHM. Consistent with this view, the 25-hep $k_{cat}$: 2-hep $k_{cat}$ ratio for the actin-activated myosin ATPase activity of I457T HMM (0.56 ± 0.09; mean + SEM for two biological replicates with three technical replicates) was nearly identical to that for the WT HMM (0.57 ± 0.03) (Fig. 7d). In the single-turnover mant-ATP experiment, the fraction of myosin in the slow phase (SRX) was 18 ± 1% for I457T 2-hep HMM and 44 ± 4% for I457T 25-hep HMM (Fig. 7c). Though the 25-hep HMM value was lower than the 59 ± 7% slow phase kinetics observed for WT type 25-hep HMM, it was closer to WT 25-hep HMM behavior that any of the four mutant 25-hep HMMs described above, consistent with more heads being in the closed IHM state for I457T 25-hep HMM than any of the other four mutants analyzed.

Since I457T has little effect on reducing the level of SRX, we sought to answer how this mutation leads to hypercontractility. Since this mutation is in the transducer region of the myosin head (Fig. 7a, b), we examined its effects on actin-activated ATPase and velocity in the in vitro motility assay of the human β-cardiac 2-hep HMM. The actin-activated ATPase $k_{cat}$ (4.2 ± 0.3 s$^{-1}$) for I457T 2-hep HMM was significantly higher than that of the WT 2-hep HMM (2.4 ± 0.06 s$^{-1}$) (Fig. 7e, Supplementary Fig. 3). The

actin gliding velocity of the I457T human β-cardiac 2-hep HMM (1700 nm s$^{-1}$) was also much higher than that of the WT 2-hep HMM (700 nm s$^{-1}$) (Fig. 7f, Supplementary Movies 1 and 2). These results suggest that the I457T mutation directly increases these activities of the myosin catalytic head, thereby leading to hypercontractility.

**Discussion**

For the past several decades there has been considerable effort to understand how HCM mutations can alter myosin power generation at the molecular level. Earlier studies using mouse models have yielded variable results[9,13]. Work investigating adult-onset HCM mutations in human β-cardiac sS1 myosin has shown mostly very small changes in ATPase activity, velocity and intrinsic force[18–20] (see Supplementary Table 5). However, while these biomechanical changes may be important for disease progression, these parameters alone are unlikely to tell the complete story. To truly understand the effect of these mutations on myosin ensemble force, we also need to account for the number of available heads to form acto-myosin crossbridges.

Previous studies have shown that myosin heads in the sarcomere can exist in a folded-back sequestered state[29,31,33]. The number of heads in this state can be modulated by the phosphorylation of RLC[30,45]. Recent biochemical binding studies have shown that HCM mutations on the myosin mesa weaken the interaction between the myosin S1 head and proximal S2 (see Adhikari et al. and Nag et al.[17,25]). These results suggest that HCM mutations that lie at the S1-S2 interface may release the sequestered heads, thereby making more heads available to form acto-myosin crossbridges, leading to increased contractility. Moreover, structural models suggest that we can apply the same reasoning to mutations that are at the PHHIS-converter S1–S1 interaction surface[20].

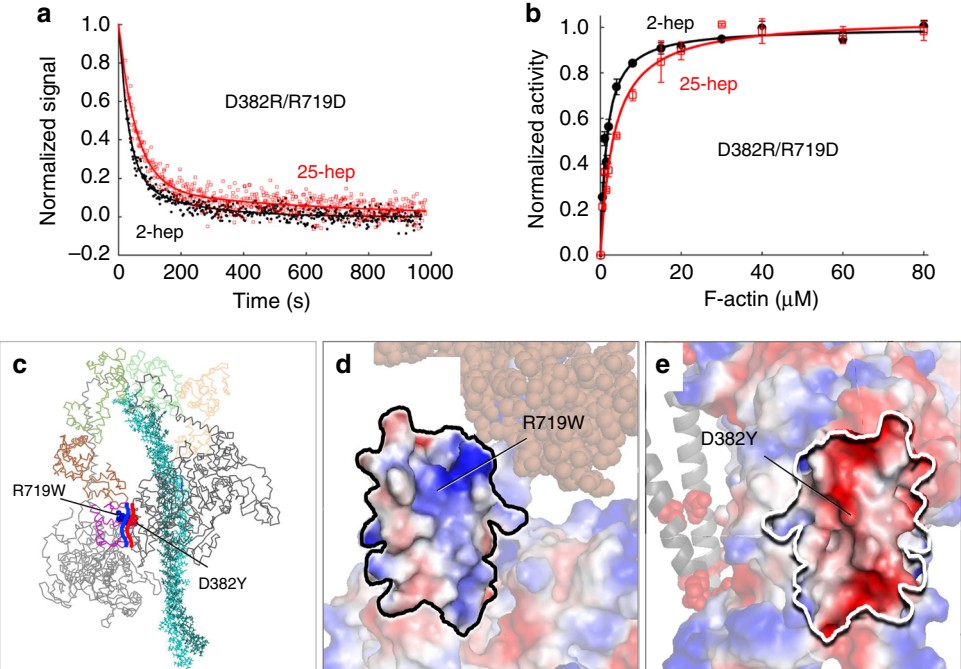

**Fig. 6** Effect of double switch mutation D382R/R719D on HMM ATPase kinetics. In all cases, black solid line and filled circles are 2-hep HMM data; Red solid line and empty squares are 25-hep HMM data. **a** Mant-nucleotide single turnover kinetics for D382R/R719D 2-hep and 25-hep HMM. **b** Actin-activated ATPase data for D382R/R719D 2-hep and 25-hep HMM. **a** Shows representative data from one preparation each. Panel **b** is combined from two independent preparations of HMMs with three experimental replicates for each preparation. Each point is an average with the error bar representing the SEM. **c** Model of the front-side view of the human β-cardiac myosin IHM (MS03 homology model, downloadable at https://spudlab.stanford.edu/homology-models/)[26], with the two S1 heads and the light chains shown as lines, and the S2 tail region represented by sticks. The Arg719 residue (blue) is part of the free head converter (purple) and the Asp382 (red) residue is part of the blocked head PHHIS. The heavy blue curved line indicates the generally positive surface of the free head converter that interacts with the generally negative blocked head PHHIS surface, marked by the heavy red curved line. **d** The image in **c** rotated 90° counterclockwise about the vertical axis defining the binding interface. The free head converter-binding interface, shown in vacuum-electrostatics mode in PyMOL, is generally positively charged. **e** The image in **c** rotated 90° clockwise about the vertical axis defining the binding interface. The blocked head PHHIS-binding interface, shown in vacuum-electrostatics mode in PyMOL, is generally negatively charged. Source data are provided as a Source Data file

However, there were no functional studies to support the hypothesis that these HCM mutations reduce the level of sequestered heads. The results reported here show four HCM mutations that do indeed decrease the level of SRX and make more heads available for interaction with actin. It is important to note that we have seen increases in functionally accessible heads for mutations that lie on the PHHIS-converter S1-S1 (D382Y, R719W) and the S1-S2 (R249Q, H251N) interfaces, but not for a control mutation that is buried in the core of the catalytic domain (I457T). These results are supported by molecular dynamics simulations that show that R249Q, H251N, D382Y, and R719W are mutations that can disrupt the IHM, thereby releasing myosin heads[28], and by a molecular modeling study of the myosin IHM which suggests that R719W is essential for the intramolecular interactions, whereas I457T is not[27]. However, I457T does show a gain in function of the catalytic head domain itself, with significant increases in the actin-activated ATPase rates and actin gliding velocity. In fact, it is important to note that the changes reported here for I457T are the largest percentage changes we have observed for any myosin HCM mutation at the molecular level. Thus, some mutations may primarily affect catalytic and biomechanical parameters (I457T) or the level of sequestered heads (R719W), while others may affect both (H251N). R719W human β-cardiac myosin sS1 showed only small changes in the biomechanical properties compared to WT myosin sS1 and we could not account for the hypercontractility seen clinically by the changes we saw in ATPase, velocity or intrinsic force[20]. H251N human β-cardiac myosin sS1, on the other hand, showed significant increases in ATPase, velocity, and intrinsic

force[17], all of which, when added to the decrease in the levels of the sequestered SRX state reported here, undoubtedly contribute to the hypercontractility seen in patients carrying these mutations.

The finding that the 25-hep HMM under the conditions studied are not 100% in the SRX state suggests that the 25-hep HMM molecules are in various states. One possibility is that 42% of the molecules are in the IHM configuration with both heads in the SRX state and the other 58% are in a fully open state with both heads functional. Alternatively, there could be some mixture of HMM molecules with both heads folded back, only one head folded back, and both heads in an open state. A third possibility is that the free head of the IHM state is indeed free to interact with actin or is pulled out of the sequestered state by an actin interaction. Further experiments are required to distinguish between these various possibilities.

Together, these data suggest that to determine how HCM mutations in human β-cardiac myosin lead to hypercontractility, one must investigate two different functional facets – (1) how the mutations alter the catalytic activity and biomechanical properties of the myosin head, and (2) how the mutations alter the intramolecular myosin interactions keeping heads in a folded-back sequestered state.

## Methods

**Expression and purification of protein.** Recombinant human β-cardiac myosin 2-hep HMM and 25-hep HMM constructs include the MYH7 gene encoding the head and either the 1st 2 or 25 heptads of the proximal S2 tail, followed by a GCN4 sequence to ensure dimerization, a flexible glycine-serine-glycine (GSG)

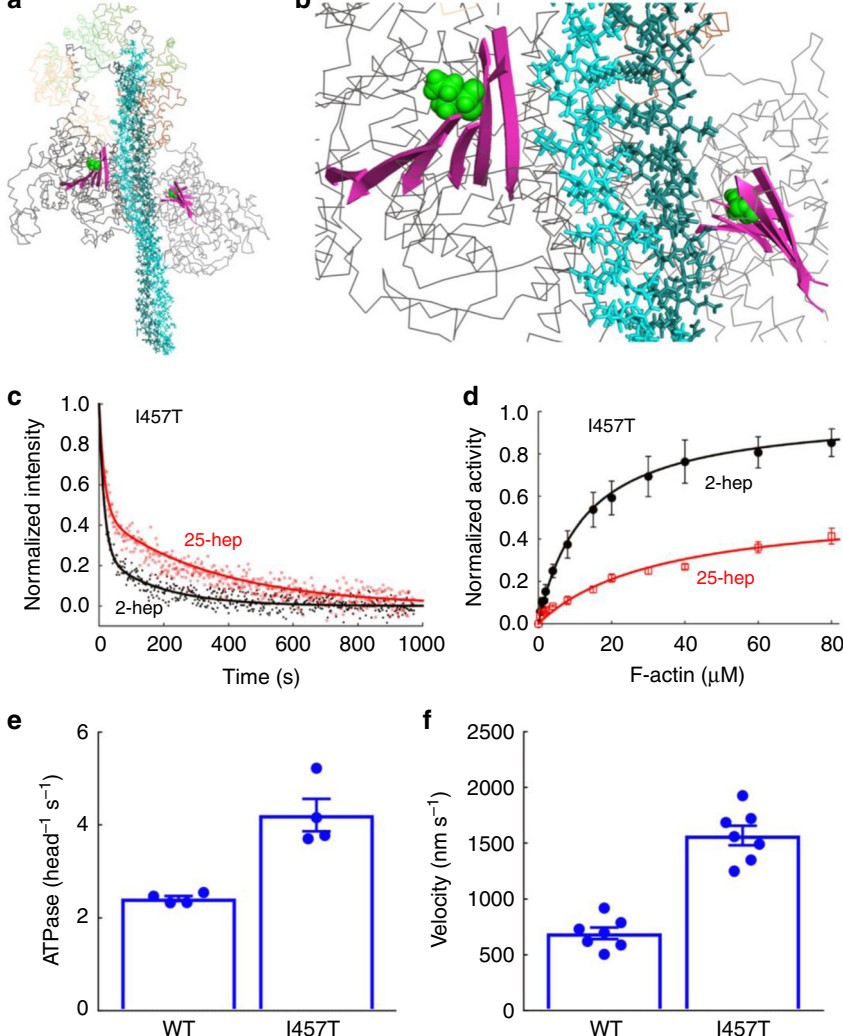

**Fig. 7** Effect of I457T mutation on HMM ATPase kinetics and in vitro motility velocity. The I457T mutation (green spheres) shown on the MS03 homology model[26] of the IHM. The mutation is located on the transducer region (pink ribbons) and is not close to the proposed interfaces within the IHM. **b** Close up of transducer region from panel **a**. **c** Mant-nucleotide release kinetics for I457T 2-hep and 25-hep HMM. **d** Actin-activated ATPase activity of I457T 2-hep and 25-hep HMM. **e** Bar plot showing the actin-activated ATPase activity of WT and I457T 2-hep HMM. **f** Bar plot showing the actin gliding velocity as measured by the in vitro motility assay of WT and I457T 2-hep HMM. In panels **e** andand **f** individual data points are denoted by closed circles. **c** Shows representative data from one preparation of each protein. Panel d is combined data from two independent protein preparations with three experimental replicates for each preparation. For panels **e** and **f**, each point is an average of two independent experiments (each from a different myosin preparation). Error bars represent the SEM. Source data are provided as a Source Data file

linker, eGFP, a second GSG linker, and finally an eight amino acid peptide (RGSIDTWV) to facilitate surface attachment via an engineered PDZ protein (see Supplementary Methods for more detail)[19]. These constructs containing the HCM mutations (mutagenic primer sequences provided in Supplementary Methods) were co-expressed with FLAG-tagged human ELC in differentiated mouse myoblast C2C12 cells (ATCC) using adenovirus vectors. After harvesting and lysing the C2C12 cells, the HMM was first bound to a FLAG-affinity resin. Then the endogenous mouse skeletal RLC, which was bound to the human cardiac HMM, was exchanged with human cardiac RLC as described in Supplementary Methods[25]. The fully assembled 2-hep and 25-hep HMMs were then purified using FLAG-affinity and anion-exchange chromatography.

Actin from bovine cardiac ventricle was a generous gift from Myokardia Inc., and was cycled through two rounds of depolymerization–repolymerization prior to use[46]. To form F-Actin for actin-activated ATPase assays, the actin was extensively dialyzed in ATPase buffer (10 mM imidazole, pH 7.5, 5 mM KCl, 1 mM DTT, and 3 mM MgCl$_2$).

**Mant-ATP single turnover**. Single turnover experiments were performed in a fluorescence plate reader (Tecan model – Infinite M200 PRO). Experiments were performed with the WT and mutant versions of human β-cardiac 25-hep HMM and 2-hep HMM in a 96-well plate (Greiner polypropylene microplate) by mixing 100 nM myosin in a buffer containing 10 mM Tris pH 7.5, 4 mM MgCl$_2$, 1 mM

EDTA, 1 mM DTT, and 5 mM potassium acetate with 2′-(or-3′)-O-(N-Methyl-lanthraniloyl) adenosine 5′-triphosphate (mant-ATP, Thermo Fischer Scientific) at a final concentration of 100 nM[24]. After 10 s, 2 mM ATP was added, followed by measuring the fluorescence signal at 470 nm after excitation at 405 nm. Fluorescence was recorded every ~2 s for 16 min total and the traces were normalized and plotted[24]. The kinetic traces were fitted to a bi-exponential decay function which yielded the amplitudes and rates of the fast (DRX rate) and slow (SRX rate) phases.

**Actin-activated ATPase assay**. Steady-state actin-activated ATPase assays of the 2-hep and 25-hep unphosphorylated HMMs were performed simultaneously for the WT and each of the mutations, with the 2-hep HMM and 25-hep HMM being expressed and purified at the same time. This assures the best comparison between the 2-hep and 25-hep HMMs since slight differences are seen from prep to prep. We used a colorimetric assay to measure inorganic phosphate production at various time points from a solution containing either 2-hep or 25-hep HMM (0.01 mg ml$^{-1}$), 2 mM ATP, and increasing amounts of actin filaments (0–80 μM). The maximum actin-activated ATPase activity ($k_{cat}$) for each measurement was calculated by fitting the data to the Michaelis–Menten equation using the curve-fitting toolbox in MATLAB. All experiments were performed at 23 °C. For each mutation, for both 2-hep and 25-hep HMM, the experiment was repeated with 2–5 different protein preparations, and for each protein preparation, three experimental

replicates were done. Representative data and $k_{cat}$ and $K_M$ values are presented in the supporting information.

**In vitro motility assay**. Multi-channel flow chambers were constructed by mounting coverslips pre-coated with 0.1% nitrocellulose/0.1% collodion dissolved in amyl acetate on a glass slide. After coating the surface first with 2 μM PDZ18 (for specific binding of HMM to the surface) and then 1 mg ml$^{-1}$ BSA to wash out unbound PDZ18 and block for non-specific binding, WT or mutant 2-hep HMM containing a C-terminal affinity tag (RGSIDTWV), which can be recognized by PDZ18 on the surface for immobilization, was flowed in. Finally, 1–5 nM bovine F-actin labeled with tetramethylrhodamine (TMR)-phalloidin, 2 mM ATP, and an oxygen scavenging system (0.4% glucose, 0.216 mg ml$^{-1}$ glucose oxidase, and 0.036 mg ml$^{-1}$ catalase) was flowed in and time-lapse images were taken on a total internal reflection fluorescence microscope coupled with a ×100 objective and an EMCCD camera at 1 Hz with 300 ms exposure. At least three movies with a duration of 30–60 s were recorded for each channel. All experiments were conducted at 23 °C. Filament tracking was performed using FAST (Fast Automated Spud Trekker) and velocities reported are the top 5% velocities[10].

**Microscale thermophoresis**. To study if mutations at the S1–S1 interface can alter S2 binding, we used MST to analyze binding between 2-hep HMM and proximal S2. We used freshly prepared 2-hep HMM (WT, R719W and D382Y) which has a C-terminal eGFP and titrated it with increasing concentrations of unlabeled proximal S2 (amino acids 839–968). Both proteins were first dialyzed into MST buffer (10 mM imidazole [pH 7.5], 100 mM KCl, 1 mM EDTA, 2 mM MgCl$_2$, 1 mM DTT, 500 mM ADP, and 0.05% Tween). The samples were loaded into NT.115 premium treated capillaries and the affinity measurements were performed at 23 °C using Nanotemper thermophoresis apparatus at MST power = 60[25]. The 2-hep HMM-S2 interaction was followed by monitoring the eGFP fluorescence. A blue LED at 30% excitation power (BLUE filter; excitation 460–480 nm, emission 515–530 nm) and IR-Laser power at 60% was used. The isotherms were fitted with both the NanoTemper commercial software and Matlab, with the Hill equation for cooperativity, to estimate the binding affinity, with a linear regression method. The binding affinities determined by both software programs were similar. Three replicates were performed for each protein preparation, and the experiments were repeated for at least 2 individual protein preparations.

**Statistical analysis**. Actin-activated ATPase assay – Each measurement at a particular actin concentration was averaged over several replicates. The error from these measurements was represented as standard error of the mean (SEM). The data were fit to the Michaelis–Menten equation to compute the $k_{cat}$ and $K_M$ values[47]. The errors on the fit, and the values were calculated using 100 bootstrap iterations.

  P-value from Student's t-test was used for testing statistical significance of the differences between 2-hep and 25-hep actin-activated ATPase rates, and fraction in slow phase derived from single ATP turnover experiments.

  Microscale thermophoresis – Student's t-test was used to determine if the mutants were statistically different from the WT.

**Reporting summary**. Further information on research design is available in the Nature Research Reporting Summary linked to this article.

## Data availability

Data supporting the findings of this manuscript are available from the corresponding authors upon reasonable request. A reporting summary for this Article is available as a Supplementary Information file.

The source data underlying Figs. 2–6, 7c–f, and Supplementary Figs. 1–3 are provided as a Source Data file.

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

## Acknowledgements

We would like to thank Ms. Kynnah Del Rosario for help with making the adenovirus, and the members of the Spudich lab for helpful discussion of the manuscript. The research was funded by NIH grants GM33289 and HL117138 (J.A.S.), a Lucile Packard CHRI fellowship (A.S.A., K.B.K., and D.V.T.), a Stanford ChEM-H Postdocs at the Interface award (to A.S.A. and K.B.K.), a Stanford CVI Postdoctoral award (to A.S.A.), an American Heart Association postdoctoral fellowship (to A.S.A. 16POST30890005; DVT 17POST33411070) and a NIH T32 Training Grant in Myocardial Biology (to K.B.K. T32 HL094274) and NIH F32HL140772 (D.S.)

## Author contributions

Investigation–Cell culture and protein preparation: A.S.A.; investigation–ATPase assays: A.S.A. and K.B.K.; investigation–single turnover: D.V.T. and S.S.S.; investigation–MST: D.V.T. and A.S.A.; investigation–in-vitro motility: D.S.; data analysis: A.S.A., D.V.T., D.S., and S.S.S.; molecular modeling: J.A.S.; initial manuscript preparation: A.S.A., J.A.S., and K.M.R.; manuscript editing: A.S.A., D.V.T., S.S.S., D.S., K.B.K., D.B., J.A.S., and K.M.R.

## Additional information

**Competing Interests:** J.A.S. is a founder of Cytokinetics and MyoKardia and a member of their advisory boards. K.M.R. is a member of the MyoKardia scientific advisory board. The remaining authors declare no competing interests.

