## [Peer Review File · Nature Communications]

Reviewers' Comments:

Reviewer #1:

Remarks to the Author:

The authors deliver here a new set of results to describe the molecular consequences of HCM mutations. Their goal is to investigate how these mutations destabilize the sequestered state and whether this would be directly linked to higher motor activity. The authors use the last modeling of this state to carefully select different kind of mutations: two mutations that are predicted to destabilize the S1-S2 interface (R249Q and H251N); two mutations predicted to destabilize the S1-S1 interface (R719W and D382Y) and one mutation (I457T) buried in the motor domain but with no direct predicted effect on the interfaces of the sequestered state. Combining structural analysis of the current structural models of the IHM and kinetic experiments such as Mant-ATP single turnover experiments and actin-activated ATPase activity measurements, the authors provide for the first time a direct evidence that single-point mutations responsible of HCM pathologies directly destabilize the IHM, leading to an increased number of available heads to produce force. This study directly demonstrate that HCM mutations can have effects on the sequestered state stability and investigate using the current IHM model detailed understanding of the bonds that form to stabilize it. The community will take benefit of this work that brings light on a fundamental concept of HCM development that was until then not formally demonstrated.

I have a few remarks on the work and few minor comments:

(1) l. 75-76: 'We have seen in most cases (except for early onset HCM mutations) that there are only very small changes compared to WT sS1, and not all parameter changes are a gain of function'. It would be interesting that the authors describe what they consider as a 'very small change' and perhaps gather in a Supplementary Table which parameter changes have been measured for the mutations they speak about. The notion of 'small change' has to be clarified since it is not easy at present to conclude about the consequences for the heart of the mutations that would result in some increase/decrease in force production. This is especially true for a disease, such as HCM, that develops during many years and that involves a different proportion of mutant/WT ratios amongst distinct cardiomyocytes.

(2) Actin-activated ATPase: the authors provide the kcat parameters in the Table S1, it would be useful to get the KM too. Some mutations seems to have some effect on this parameter, it is the case for example for H251N that seems to have a higher KM than R249Q. If this is true, it would be interesting to discuss the effect of the mutations on this parameter too. More generally, the study would gain in accuracy if the authors discussed briefly the differences in kinetics observed in mutations that are close in space (R249Q and H521N for example which have quite different effects in terms of magnitude on the activity).

(3) l. 250 – 'there is a net positive charge on the free head converter surface, which interacts with an overall positively-charged surface on the blocked head' - are the two surfaces really positively charged? This should be illustrated in a figure.

(4) l. 242-259: the authors should moderate their conclusions regarding the results of the double-mutant. R719 is altering the dynamics of the so-called top-loop, which is predicted to be a part of the interface between the blocked head and the free-head (Robert-Paganin et al., 2018). It cannot be excluded that the double-mutant could have long-range effects on the conformation and dynamics of the top-loop that would alter the ability of each head to interact with each other. The authors could also take this into account in their discussion.

(5) In discussions: the authors should not minimize the fact that mutations can have effects on both the mechano-chemical parameters of the myosin and the sequestered state, as it is shown by the comparison of ATPase parameters of the 2-hept between the WT and all the mutants studied here. Which is consistent with previous hypotheses. It would be important to mention in discussion that some mutants have effects on both the stability of the sequestered state and motor activity.

Minor comments:

- (1) On the Microthermophoresis experiments: the authors should provide the numerical values of the K_d computed from the experiments.
- (2) Bibliography: the reference 28 is now published in Nat Comms (<https://doi.org/10.1038/s41467-018-06191-4>).
- (3) Figure 6: on this figure, the authors present results on the double-mutant R719D/D389R, but the label on the figure is R719W/D382Y.
- (4) Line 280 typo : I475T is written instead of I457T

Reviewer #2:

Remarks to the Author:

The authors report an analysis of the effects of four HCM mutations located at the head-tail (R249Q,H251N) and head-head (D382Y,R719W) interface of the interacting heads motif on the enzymatic activity of beta-cardiac myosin. They can effectively evaluate the effect of these interactions by comparing two HMM constructs, 25-hep HMM and 2-hep HMM, the latter missing repeats of the proximal S2 tail that was previously demonstrated to be essential for the interaction and regulation of beta-cardiac myosin by RLC phosphorylation. The myosin enzymatic activity is measured by mant-ATP single turnover assay, which allows the authors to evaluate the percentage of myosins showing slow (SRX) and fast ATP-ase turnover rates, as well as by actin-activated ATPase assay.

In previous work (ref 25,17), the authors demonstrated that some HCM mutations lead to reduction in the head-tail interaction and that a reduction in the head-tail interaction leads to increased enzymatic activity of cardiac myosin, thus indicating (but not demonstrating) that HCM mutations lead to increased enzymatic activity. This research work is a considerable effort in producing recombinant constructs to compare the properties of beta-cardiac myosins carrying different hypertrophic cardiomyopathy mutations (HCM). Despite some points listed below, the experimental work is technically sound and the authors convincingly demonstrate that HCM mutations release the sequestered myosin heads and increase the percentage of enzymatically active heads. Although this is a step forward in understanding the effect of HCM mutations on hypertrophic cardiomyopathies, my opinion is that the article in the present form would be more appropriate for a more specialized journal. One important point that is missing in the paper is a report of the mechanical activity of HCM myosin. This is a fundamental point because the ATPase rate is not the only factor affecting contractility and motor proteins with higher ATPase can sometimes show slower velocity and force. Experimental evidence of a link between HCM mutations and increased mechanical output of the motor (for example measuring the sliding velocity of HCM mutants in a gliding assay) could greatly extend the importance of the results. Moreover, the last sentence in the abstract, which states that HCM mutations lead to "hypercontractility at the molecular level" is not supported by the ATPase data presented here but would need mechanical data.

Other major points:

1. The working principle of MST experiments should be briefly described in the text for a general reader. Importantly, the fitting curve in Fig. 1 is not described and fitting parameters with statistical errors are not reported. Therefore I do not see how the authors' statement on lines 165-166 can be derived from Fig. 1 ("the observed affinities represented less than a 2-fold change compared to the WT 2-hep HMM-proximal S2 affinity")
2. All fits in Fig. 3, 4, 5, 6, and 7 are missing some values of the fit parameters and errors:
 - Only the percentage of fast and slow phases in mant-ATP experiments is reported (parameters a_1 and a_2 in the equation on line 192), but not the rates b_1 and b_2 . Do also rates b_1 and b_2 change between experiments and, if so, why?
 - Only k_{cat} is reported in the ATPase assay. What is the value of K_M , does it varies between

experiments and, if so, why? (for example, KM for H251N 2-hep and 25-hep reported in fig. 4d look different)

Minor points:

3. the title of the manuscript states that HCM mutations "increase myosin enzymatic activity". The authors demonstrate that HCM mutations lead to increased percentage of enzymatically active heads, but the absolute value of the 2-hep HMM kcat is smaller for the R249Q mutation compared to WT (Table S1). Is the increased kcat of the 25-hep HMM sufficient to have an increased average enzymatic activity of the R249Q mutation compared to WT?
4. lines 172-173: "We used the 2-hep HMM as a control because it shows the same maximal actin-activated ATPase activity as sS1 alone". This is not shown in the manuscript.
5. lines 196, 200: What are the errors reported for a1 and a2 (S.D., S.E., 95% C.I., ...)?
6. lines 211-213: What are the errors reported for kcat?
7. When making comparisons between parameters, please give the p-value and statistical test used.
8. Table S1: What is the "D/R" column?
9. The supplemental methods describing the expression of the constructs lack details about how the HCM mutations were introduced in the constructs.

Reviewer #3:

Remarks to the Author:

Summary: The study examines the importance of the ability of an HCM mutations effect on the formation of the inhibited conformation, usually called the interacting heads motif, to affect biochemical properties of the human β -myosin II. The study examines four mutations in regions that would be predicted to affect interacting heads motif formation and one that would be predicted to have no effect on formation, but which would affect myosin's enzymatic properties. The study is well done and thorough.

Critique: This is an important paper that should advance how these mutations in cardiac myosin are studied and thus increase the prospects that a therapy might finally be obtained. The study examines a number of properties of these mutations within the context of the double headed myosin construct, whereas typically they are examined using single myosin head constructs or fragments obtained by proteolysis. Certainly biochemists should be examining the 2-headed myosin fragments because that is the most functional construct.

The study is thorough. They examined two constructs at the interface between myosin heads in the inactive conformation and two constructs on the myosin mesa plus one construct having a mutation predicted based on current knowledge that could not have anything to do with these two interfaces. In the latter case there was no significant difference in the binding of S2 peptide fragments to the myosin heads and little effect on the ability of the inhibited conformation to form.

I only have minor comments which are made below which the authors can feel free to ignore. Altogether it is a very nice piece of work that certainly meets the quality .

Minor points:

I prefer the term MDE instead of sS1. MDE was used for the first crystal structure of smooth muscle myosin (Dominguez et al.) and since that conformation is very similar to that of the blocked head in the smooth muscle IHM, I would argue it is a somewhat more descriptive and more readily "assimilated" acronym for those in the muscle field. Just a suggestion, not a request.

What is the typical experimental uncertainty of the microscale thermophoresis in general? If the <

2x change measured here (lines 163-166) is within the typical accuracy of the technique, it is worth mentioning this fact to the general audience.

Reviewers' comments:

Reviewer #1 (Remarks to the Author):

The authors deliver here a new set of results to describe the molecular consequences of HCM mutations. Their goal is to investigate how these mutations destabilize the sequestered state and whether this would be directly linked to higher motor activity. The authors use the last modeling of this state to carefully select different kind of mutations: two mutations that are predicted to destabilize the S1-S2 interface (R249Q and H251N); two mutations predicted to destabilize the S1-S1 interface (R719W and D382Y) and one mutation (I457T) buried in the motor domain but with no direct predicted effect on the interfaces of the sequestered state. Combining structural analysis of the current structural models of the IHM and kinetic experiments such as Mant-ATP single turnover experiments and actin-activated ATPase activity measurements, the authors provide for the first time a direct evidence that single-point mutations responsible of HCM pathologies directly destabilize the IHM, leading to an increased number of available heads to produce force. This study directly demonstrate that HCM mutations can have effects on the sequestered state stability and investigate using the current IHM model detailed understanding of the bonds that form to stabilize it. The community will take benefit of this work that brings light on a fundamental concept of HCM development that was until then not formally demonstrated.

I have a few remarks on the work and few minor comments:

(1) l. 75-76: 'We have seen in most cases (except for early onset HCM mutations) that there are only very small changes compared to WT sS1, and not all parameter changes are a gain of function'. It would be interesting that the authors describe what they consider as a 'very small change' and perhaps gather in a Supplementary Table which parameter changes have been measured for the mutations they speak about. The notion of 'small change' has to be clarified since it is not easy at present to conclude about the consequences for the heart of the mutations that would result in some increase/decrease in force production. This is especially true for a disease, such as HCM, that develops during many years and that involves a different proportion of mutant/WT ratios amongst distinct cardiomyocytes.

---We thank the reviewer for the comment, and we agree that it is difficult to predict what level of change in our biochemical and biomechanical parameters is required to cause HCM over time, especially given the phenotypic heterogeneity seen even among family members with the same mutation. Our approach to this question initially was to look for changes in the basic biochemical and biomechanical parameters, hoping to find changes in ATPase, velocity and/or intrinsic force in the >20-50% range. This was in fact what we observed for the pediatric-onset mutations, as may be expected given their more severe phenotype. However, almost all of the adult-onset mutations have shown smaller (<10-20%) changes, and often have some parameters that change in the direction of loss of function, making it difficult to argue that the overall effects on these fundamental parameters were a hypercontractile response; indeed two adult onset mutations gave 0% change in all three of these parameters! These observations are what pushed us to look for other factors we might be missing, such as N_a . We have clarified in the text what we consider to be a small change (~10-20%) and now include a table in the supplemental information (Table S3) where we include the mechano-chemical parameters (intrinsic force, ATPase activity, and actin gliding velocity) for all the mutations we have investigated using the sS1 myosin. Accompanying the table is a

discussion of the changes that are observed for the mutations that have not previously been published.

(2) Actin-activated ATPase: the authors provide the k_{cat} parameters in the Table S1, it would be useful to get the K_M too. Some mutations seems to have some effect on this parameter, it is the case for example for H251N that seems to have a higher K_M than R249Q. If this is true, it would be interesting to discuss the effect of the mutations on this parameter too. More generally, the study would gain in accuracy if the authors discussed briefly the differences in kinetics observed in mutations that are close in space (R249Q and H521N for example which have quite different effects in terms of magnitude on the activity).

--- We thank the reviewer for the suggestion. We have now included the K_M values for all the mutations tested in Table S1. In general, we see that the K_M values for the mutants are for the most part within 2-fold of WT (in either direction), with I457T in the transducer being the outlier with a ~3.5 fold increase in K_M . None of the mutations studied are near the actin-binding interface; however, given the highly allosteric nature of the myosin motor, it is certainly possible that changes such as I457T in the transducer affect the apparent affinity of the motor for actin. A recent study from the Geeves' laboratory (Ujfalusi et al JBC 2018 doi: 10.1074/jbc.RA118.001938) showed that for a variety of HCM- and DCM-causing mutations in β -cardiac sS1, K_M values varied as much as two-fold or more and were not predictive of disease phenotype.

We agree that the difference in activity between R249Q and H251N is interesting, given the proximity of the two mutated residues. Perhaps the loss of the positively charged arginine side chain adversely impacts the transducer and leads to decreased ATPase activity. Such questions almost certainly would need structural studies to provide satisfactory answers. The K_M values reported in Table S1 for R249Q and H251N are essentially identical for the 2hep versions of these mutations but are significantly different for the 25hep versions, as the reviewer points out. R249Q is an outlier in this respect – its 2hep and 25hep motors have similar K_M values, whereas the other 25hep motors have ~2-3-fold higher K_M values than those of their 2-hep counterparts. The reason for this is not entirely clear, and we are beginning to explore whether the presence of the leucine zipper after only 2 heptad repeats of the proximal S2 may affect the interaction of the two heads with the actin filament. As these studies necessitate the creation of new constructs (a many month process given our adenovirus-based expression system) such experiments are beyond the scope of this study.

(3) I. 250 – ‘ there is a net positive charge on the free head converter surface, which interacts with an overall positively-charged surface on the blocked head’ - are the two surfaces really positively charged? This should be illustrated in a figure.

--- We thank the reviewer for pointing out this typo, and we have corrected the manuscript to read “which interacts with the overall negatively-charged surface on the blocked head.” We have also added panels to Fig. 6 to illustrate the interface of the positively and negatively charged surfaces.

(4) I. 242-259: the authors should moderate their conclusions regarding the results of the double-mutant. R719 is altering the dynamics of the so-called top-loop, which is predicted to be a part of the interface between the blocked head and the free-head (Robert-Paganin et al., 2018). It cannot be excluded that the double-mutant could have long-range effects on the conformation and dynamics of the top-loop that would alter the ability of each head to interact with each other. The authors could also take this into account in their discussion.

--- We thank the reviewer for the insight and agree completely. We have altered our discussion to address this point accordingly.

(5) In discussions: the authors should not minimize the fact that mutations can have effects on both the mechano-chemical parameters of the myosin and the sequestered state, as it is shown by the comparison of ATPase parameters of the 2-hep between the WT and all the mutants studied here. Which is consistent with previous hypotheses. It would be important to mention in discussion that some mutants have effects on both the stability of the sequestered state and motor activity.

--- We agree with the reviewer's point and did not mean to minimize the idea that effects on mechano-chemical parameters also clearly play a role – that is clearly illustrated by the I457T mutation. We have modified our discussion accordingly. We had previously mentioned that we need to account for the changes in catalytic activity and sequestered state to gain a complete understanding of how a mutation can alter myosin function. Now we have also included examples of H251N and R249Q to illustrate the point, and moreover, also refer to the table of fractional changes suggested by the reviewer so that the reader can gain a better insight of the heterogeneity in the catalytic changes of myosin due to these mutations.

Minor comments:

(1) On the Microthermophoresis experiments: the authors should provide the numerical values of the K_d computed from the experiments.

--- These values have now been incorporated into the Results section.

(2) Bibliography: the reference 28 is now published in Nat Comms (<https://doi.org/10.1038/s41467-018-06191-4>).

--- This reference has been updated.

(3) Figure 6: on this figure, the authors present results on the double-mutant R719D/D389R, but the label on the figure is R719W/D382Y.

--- We thank the reviewer for catching this mistake. It has been fixed.

(4) Line 280 typo : I475T is written instead of I457T

--- We thank the reviewer for pointing out the typo. We have fixed it.

Reviewer #2 (Remarks to the Author):

The authors report an analysis of the effects of four HCM mutations located at the head-tail (R249Q,H251N) and head-head (D382Y,R719W) interface of the interacting heads motif on the enzymatic activity of beta-cardiac myosin. They can effectively evaluate the effect of these interactions by comparing two HMM constructs, 25-hep HMM and 2-hep HMM, the latter missing repeats of the proximal S2 tail that was previously demonstrated to be essential for the interaction and regulation of beta-cardiac myosin by RLC phosphorylation. The myosin enzymatic activity is measured by mant-ATP single turnover assay, which allows the authors to evaluate the percentage of myosins showing slow (SRX) and fast ATP-ase turnover rates, as well as by actin-activated ATPase assay.

In previous work (ref 25,17), the authors demonstrated that some HCM mutations lead to reduction in the head-tail interaction and that a reduction in the head-tail interaction leads to increased enzymatic activity of cardiac myosin, thus indicating (but not demonstrating) that HCM mutations lead to increased enzymatic activity. This research work is a considerable effort in producing recombinant constructs to compare the properties of beta-cardiac myosins carrying different hypertrophic cardiomyopathy mutations (HCM). Despite some points listed below, the experimental work is technically sound and the authors convincingly demonstrate that HCM mutations release the sequestered myosin heads and increase the percentage of enzymatically active heads. Although this is a step forward in understanding the effect of HCM mutations on hypertrophic cardiomyopathies, my opinion is that the article in the present form would be more appropriate for a more specialized journal. **One important point that is missing in the paper is a report of the mechanical activity of HCM myosin. This is a fundamental point because the ATPase rate is not the only factor affecting contractility and motor proteins with higher ATPase can sometimes show slower velocity and force.** Experimental evidence of a link between HCM mutations and increased mechanical output of the motor (for example measuring the sliding velocity of HCM mutants in a gliding assay) could greatly extend the importance of the results. Moreover, the last sentence in the abstract, which states that HCM mutations lead to “hypercontractility at the molecular level” is not supported by the ATPase data presented here but would need mechanical data.

We agree fully with the reviewer that all the parameters have to be taken into consideration when exploring the causes of hypercontractility studying these purified proteins. The mechanical activity of the R719W mutant myosin studied here was reported in Kawana, M., Sarkar, S.S., Sutton, S., Ruppel, K.M. & Spudich, J.A. Biophysical properties of human beta-cardiac myosin with converter mutations that cause hypertrophic cardiomyopathy. Sci Adv 3, e1601959 (2017). In that case, the results surprised us in that there were only small changes in the biomechanical properties, and we could not account for the hypercontractility seen clinically by the changes we saw in ATPase, velocity or intrinsic force. As stated in the introduction of the current paper, this has been true of the majority of adult onset HCM mutations we have studied, which led us to propose that release from a sequestered state may be the more common effect of HCM mutations, and we are currently working to assess that hypothesis. The mechanical activities of the H251N mutant myosin studied here were reported in Adhikari, A.S. et al. Early-Onset Hypertrophic Cardiomyopathy Mutations Significantly Increase the Velocity, Force, and Actin-Activated ATPase Activity of Human β -Cardiac Myosin. Cell Reports 17, 2857-2864 (2016). This mutation does in fact give

increases in ATPase, velocity and intrinsic force, all of which, when added to the decrease in the levels of the sequestered SRX state reported here, undoubtedly contribute to the hypercontractility seen in patients carrying this mutation. These considerations have now been added to the Conclusion.

We would like to note that we carried out some motility experiments with these 25hep mutants compared to WT25hep to see if we could show that higher concentrations of WT myosin compared to mutant myosin may be needed for maximum velocity (since the WT myosin has more sequestered SRX heads that are not in play). As we suspected, this approach is not sensitive enough to detect the ~2-fold difference in concentrations that we might have suspected; furthermore, this assay involves putting the myosin on a surface, which may well alter the equilibrium between folded-back and open states. In vitro motility therefore is not an appropriate assay for measuring changes in the amount of sequestered off-state myosin.

Other major points:

1. The working principle of MST experiments should be briefly described in the text for a general reader. Importantly, the fitting curve in Fig. 1 is not described and fitting parameters with statistical errors are not reported. Therefore I do not see how the authors' statement on lines 165-166 can be derived from Fig. 1 ("the observed affinities represented less than a 2-fold change compared to the WT 2-hep HMM-proximal S2 affinity")

--- We thank the reviewer for pointing this omission. We have now added a brief description of MST methodology and added details of fitting parameters and statistical analysis for Figure 1.

2. All fits in Fig. 3, 4, 5, 6, and 7 are missing some values of the fit parameters and errors:
- Only the percentage of fast and slow phases in mant-ATP experiments is reported (parameters a1 and a2 in the equation on line 192), but not the rates b1 and b2. Do also rates b1 and b2 change between experiments and, if so, why?

--- We thank the reviewer for the comment. For all the actin-activated ATPase data, the respective k_{cat} and K_m values with the standard error have been reported in Supplemental Table 1. The number of replicates for each measurement is reported in the figure legend. Bootstrap analysis is used to calculate the error.

For the single turnover experiments, we have now included the rates in the supplemental table S2a and S2b for 2-hep HMM and 25-hep HMM, respectively. As can be seen from the tables, the rates are consistent for the mutants, with the fast phase rate approximately an order of magnitude higher than the slow rate, consistent with the DRX and SRX rates previously published in the literature.

- Only k_{cat} is reported in the ATPase assay. What is the value of K_M , does it varies between experiments and, if so, why? (for example, K_M for H251N 2-hep and 25-hep reported in fig. 4d look different)

--- We thank the reviewer for the comment. We have now included the K_M values for all the mutations measured for both the 2-hep and 25-hep HMMs in supplemental table 1. As discussed above in response to a similar point raised by Reviewer 1, in general the 25-hep motors have ~2-3-fold higher K_M values than those of their 2-hep counterparts. The reason for this is not entirely clear, and we are beginning to explore whether the presence of the leucine zipper after only 2 heptad repeats of the proximal S2 may affect the interaction of the two heads with the actin filament. As these studies necessitate the creation of new constructs (a many month process given our adenovirus-based expression system) such experiments are beyond the scope of this study.

Minor points:

3. the title of the manuscript states that HCM mutations "increase myosin enzymatic activity". The authors demonstrate that HCM mutations lead to increased percentage of enzymatically active heads, but the absolute value of the 2-hep HMM k_{cat} is smaller for the R249Q mutation compared to WT (Table S1). Is the increased k_{cat} of the 25-hep HMM sufficient to have an increased average enzymatic activity of the R249Q mutation compared to WT?

--- For the 25-hep ATPase activity measured, the k_{cat} of R249Q is ~20% higher than the WT myosin (1.38 vs 1.63 $s^{-1}head^{-1}$, see Table S1), a statistically significant increase.

4. lines 172-173: "We used the 2-hep HMM as a control because it shows the same maximal actin-activated ATPase activity as sS1 alone". This is not shown in the manuscript.

--- We thank the reviewer for raising this point. We have now included actin activated ATPase data comparing WT sS1 and 2-hep HMM, and R719W sS1 and 2-hep HMM in the supplemental data.

5. lines 196, 200: What are the errors reported for a_1 and a_2 (S.D., S.E., 95% C.I., ...)?

--- These are standard error, and we have now stated that both in the figure legend and the methods section of the revised manuscript.

6. lines 211-213: What are the errors reported for k_{cat} ?

--- These are standard error, and we have now stated that both in the figure legend and the methods section of the revised manuscript.

7. When making comparisons between parameters, please give the p-value and statistical test used.

--- We have now included p-values as requested as well as the statistical test used.

8. Table S1: What is the "D/R" column?

--- *We apologize for the unclear abbreviation. This is the double mutation, D382R/ R719D, and we have made that clear in the revised manuscript.*

9. The supplemental methods describing the expression of the constructs lack details about how the HCM mutations were introduced in the constructs.

--- *We thank the reviewer and have now included more detailed methods to describe the site directed mutagenesis protocol used.*

Reviewer #3 (Remarks to the Author):

Summary: The study examines the importance of the ability of an HCM mutations effect on the formation of the inhibited conformation, usually called the interacting heads motif, to affect biochemical properties of the human β -myosin II. The study examines four mutations in regions that would be predicted to affect interacting heads motif formation and one that would be predicted to have no effect on formation, but which would affect myosin's enzymatic properties. The study is well done and thorough.

Critique: This is an important paper that should advance how these mutations in cardiac myosin are studied and thus increase the prospects that a therapy might finally be obtained. The study examines a number of properties of these mutations within the context of the double headed myosin construct, whereas typically they are examined using single myosin head constructs or fragments obtained by proteolysis. Certainly biochemists should be examining the 2-headed myosin fragments because that is the most functional construct.

The study is thorough. They examined two constructs at the interface between myosin heads in the inactive conformation and two constructs on the myosin mesa plus one construct having a mutation predicted based on current knowledge that could not have anything to do with these two interfaces. In the latter case there was no significant difference in the binding of S2 peptide fragments to the myosin heads and little effect on the ability of the inhibited conformation to form.

I only have minor comments which are made below which the authors can feel free to ignore. Altogether it is a very nice piece of work that certainly meets the quality .

--- *We thank the reviewer for the positive comments!*

Minor points:

I prefer the term MDE instead of sS1. MDE was used for the first crystal structure of smooth muscle myosin (Dominguez et al.) and since that conformation is very similar to that of the blocked head in the smooth muscle IHM, I would argue it is a somewhat more descriptive and more readily "assimilated" acronym for those in the muscle field. Just a suggestion, not a request.

--- *We thank the reviewer for this suggestion, and have revised the manuscript by referring to the MDE nomenclature when first introducing the construct. However, all of our previous*

published work analyzing mutations in the context of this construct have referred to it as sS1. We believe it will be less confusing to readers to continue to refer to it as sS1 in our manuscript rather than changing the nomenclature at this point.

What is the typical experimental uncertainty of the microscale thermophoresis in general? If the < 2x change measured here (lines 163-166) is within the typical accuracy of the technique, it is worth mentioning this fact to the general audience.

--- The uncertainty in affinity measured by MST varies between different experiments and different binding partners, hence it is difficult to generalize a particular value for the uncertainty. For example, in our hands, we have observed affinities for the S1 GFP/ proximal S2 interaction that vary from 30 μ M to 70 μ M (Table S1, column 5, Nag et al., NSMB, 2017). Thus, here, the reported K_d 's for R719W and D382Y are within a 2-fold change compared to the WT 2-hep HMM-proximal S2 affinity.

Reviewers' Comments:

Reviewer #1:

Remarks to the Author:

The authors have fully answered all the requests and comments. I think this contribution is important for the field and that the paper is now ready for publication.

Reviewer #2:

Remarks to the Author:

The authors addressed all my specific concerns on the manuscript other than the lack of data on the mechanical activity of the mutations. This limits the reach of the article, but not the scientific validity of the work, which is high-quality and now appropriate for publication.